# Non-Destructive Measuring Systems for the Evaluation of High Oxygen Stored Poultry: Development of Headspace Gas Composition, Sensory and Microbiological Spoilage

**DOI:** 10.3390/foods11040592

**Published:** 2022-02-18

**Authors:** Jasmin Dold, Caroline Kehr, Clarissa Hollmann, Horst-Christian Langowski

**Affiliations:** 1Chair of Brewing and Beverage Technology, Technical University of Munich, Weihenstephaner Steig 20, D-85354 Freising, Germany; caroline.kehr@tum.de (C.K.); clarissahollmann@gmail.com (C.H.); 2TUM School of Life Sciences, Technical University of Munich, Weihenstephaner Steig 22, D-85354 Freising, Germany; h-c.langowski@tum.de; 3Fraunhofer Institute for Process Engineering and Packaging, Giggenhauser Straße 35, D-85354 Freising, Germany

**Keywords:** modified atmosphere packaging, non-destructive, shelf-life prediction, sensory evaluation, fluorescence quenching, infrared spectroscopy, meat quality

## Abstract

As poultry is known to be a perishable food, the use-by date is set in such a way that food safety is guaranteed even with a higher initial bacterial count. This means, however, that some products are wasted, even if they are still safe to eat. Therefore, non-destructive measurement devices might be a good opportunity for individual shelf-life prediction, e.g., in retail. The aim of this study was therefore to use non-destructive measurement devices based on fluorescence quenching (oxygen detection) and mid-infrared laser spectroscopy (carbon dioxide detection) for the monitoring of high-oxygen-packed poultry in different storage conditions. During 15 days of storage, the gas composition of the headspace was assessed (non-destructively and destructively), while total plate count was monitored and a comprehensive sensory evaluation was performed by a trained panel. We were able to demonstrate that in most cases, non-destructive devices have comparable precision to destructive devices. For both storage conditions, the sensory attribute *slime* was correlated with reaching the critical microbiological value of 10^7^ CFU/g; the attribute *buttery* was also useful for the prediction of regularly stored poultry. The change in the gas atmosphere as a sign of premature spoilage, however, was only possible for samples stored in irregular conditions.

## 1. Introduction

Food waste is a global issue, estimated by the FAO (Food and Agriculture Organization of the United Nations) to be about 1.3 billion tons per year worldwide [1]. In the wealthier industrialized countries, more than 50% of food waste happens during distribution in the retail sector or at the consumer’s home, often due to an expired “best before” or “use by” date [1,2]. The individual quality evaluation of an already packed foodstuff represents a major research topic of our time for different reasons. First of all, the quality standards and consumer expectations of foodstuffs are constantly increasing. To integrate an intelligent sensor, e.g., a colorimetric dye, into a packaging system can serve as both a safeguard for the manufacturer to ensure quality and at the same time to give the consumer the feeling that they are in charge of quality control. The latter, however, brings the disadvantage that consumers are often insecure in estimating the information in the right way, which can cause unnecessary food waste [3]. In addition, the risk remains that the indicator is false negative due to the lack of the relevant metabolites, for example, something that could have very negative consequences if pathogens (e.g., *Campylobacter jejuni*) are present [3,4]. Contrary to the earlier assumption that chicken meat is best packed in a low-oxygen (O_2)_ atmosphere [5], nowadays, a modified atmosphere packaging (MAP) with a high O_2_ and a low carbon dioxide (CO_2_) content is often chosen, as the high O_2_ amount is known to inhibit the growth of the anaerobic pathogen *C. jejuni* [6]. Therefore, some studies focusing on the growth of microorganisms on fresh poultry under high O_2_ atmosphere were conducted. By identifying different types of microorganisms as being part of the spoilage—such as *Brochothrix thermosphacta* or *Carnobacteria*—they also noticed a decrease in O_2_ and an increase in CO_2_ in the headspace of the packaging during the storage time, measured via destructive gas detection devices [7,8]. Further studies confirmed the connection of O_2_ consumption in the headspace of packed meat products with the growth of different types of microorganisms [9,10]. However, the authors were not in full agreement whether the change in gas composition was really correlated with the critical value of spoilage or not. The critical value is defined as 10^7^ colony-forming units per gram of sample (CFU/g) [11].

The quality determination of food by optical measurement methods has some advantages that cannot be denied. First of all, most of the methods are very fast and easy to handle. In addition, a lot of successful research that provides a good basis for the comparability and interpretation of results was conducted in the past. Another big advantage is that the measurements are non-destructive in most cases and can therefore be used for the quality evaluation of already packed foodstuffs. There are systems available over a wide range of the optical spectrum, with several advantages and disadvantages, as we already discussed in Dold and Langowski (2022) [12]. In our work shown here, we used measurement systems based on fluorescence quenching and infrared (IR) spectroscopy. Fluorescence quenching—or more precisely, dynamic quenching—can be used for determining O_2_ concentrations, as these molecules are able to absorb the energy of a fluorophore excited by visible light which leads to a shorter luminescence lifetime as a consequence. This effect can be described with the Stern–Volmer equation [13]. It is therefore a useful tool for the detection of packaging integrity for modified atmosphere packed products [14] but may also be useful for a statement on microbiological spoilage [15], by successful integration into a suitable packaging system. Measurement systems based on IR are also well studied for the quality determination of food and already packaged foodstuffs. Schmutzler et al., for example, studied the adulteration of veal sausage with pork products, using different near-infrared (NIR) devices in 2016 [16]. When light is used in the NIR and mid-infrared (MIR) range, vibrational states of the molecules are excited; in the far-infrared (FIR), IR radiation leads to the excitation of rotational states [17]. The resulting absorption peaks are characteristic for many different properties, such as gas concentration [18] or moisture content [19]. In this work, we used a device working in the MIR range for the detection of CO_2_.

At the 2nd International Electronic Conference on Foods—“Future Foods and Food Technologies for a Sustainable World”, we already showed some of our results, showing that non-destructive measurement devices are able to monitor the gas atmosphere in the headspace of transparent packages. Therefore, high-O_2_-packed poultry was stored over a period of 15 days at two different temperatures. We showed the development of the headspace concentrations of O_2_ and CO_2_ during storage via the non-destructive devices and the development of the microbiological spoilage in comparison to sensory acceptability. These results are also presented in the following publication. We concluded that the systems cannot be used for the shelf-life prediction of regularly stored poultry with only these parameters. We decided that a correlation with volatile emissions (volatile organic compounds (VOCs)) might be useful, which had to be part of an extensive sensory evaluation. In this publication, we show a more extensive sensory panel evaluation of the samples during storage, as well as the results of the previous training of the panelists. These results were further correlated in detail with the gas change and microbiological spoilage. In addition, the formed VOCs were theoretically assigned to typical meat spoilers. We also studied the influence of the headspace to meat ratio for best possible measurement accuracy. To further demonstrate the suitability of the non-destructive devices, their measurement precision was compared to that of the destructive devices.

## 2. Materials and Methods

### 2.1. Panel Training for the Sensory Evaluation

For the sensory evaluation of the samples, a panel (*n* = 15) consisting of 5 female and 10 male participants with an average age of 29 years was selected.

First, a descriptive analysis was carried out. Therefore, the panelists had to describe two poultry strips in their own words. The first sample was fresh slaughtered poultry and the second a forced-aged sample (4 days stored at 23 °C). A distinction was thereby made between visual and olfactory impression.

Based on the descriptive analysis, fitting references for further training (olfactory references) and for the sensory evaluation (olfactory and visual references) were selected and prepared, respectively. As olfactory references, common odor substances (e.g., diacetyl for *buttery*) were used. For the visual evaluation, a color chart for fresh and forced-aged poultry was designed. For this purpose, the RGB values of the meat samples were measured and transferred for both fresh and forced-aged poultry to a color chart, which was available to the panelists during the evaluation. In addition, images of fresh and forced-aged poultry were printed as image reference.

Subsequently, the named attributes were collected, and the most frequently described attributes were selected for the second part of the training, consisting of an odor identification test. For this purpose, the participants were given standardized flavors (references) corresponding to the attributes selected above, which they had to match to the correct attributes. In addition, they had to evaluate the odor intensity for each sample with the help of a line scale, ranging from weak (0) to strong (100). This was used to determine how sensitive each panelist was to the flavors.

### 2.2. Determination of the Headspace Gas Atmosphere

#### 2.2.1. Non-Destructive Determination of O_2_ Concentration

For the non-destructive determination of O_2_ gas concentration, a fluorescence-based measurement system and associated sensor spots containing a fluorescent dye (PreSens Precision Sensing GmbH, Regensburg, Germany) were used. The measurement device works via fiber optics, exciting the embedded fluorophores at a wavelength of 505 nm. The excited fluorophores emit fluorescent light at a wavelength of 650 nm, when returning back to the ground state. Fiber optics has an integrated photodiode (which measures the luminescence lifetime of the emitting fluorophores) that is directly linked with the O_2_ concentration, as already described in the Introduction section. To integrate the sensor spots into the lid film, a sensor spot was placed on the inside of the lid film, (PP/PA/PP/PA, 100 μm, allvac Folien GmbH, Waltenhofen, Germany) covered with a PP film (56 μm, Huhtamaki Flexible Packaging Germany GmbH & Co. KG, Ronsberg, Germany) and sealed with a ring-shaped sealing tool at 155 °C. Before sealing, a two-point calibration of the sensor spots in the relevant measuring range (0 and 60% O_2_ % (*v*/*v*)) was carried out. All O_2_ gas concentrations mentioned in the text are to be understood as percent by volume (*v*/*v*).

#### 2.2.2. Non-Destructive Determination of CO_2_ Concentration

The non-destructive measurement of CO_2_ was carried out with a measurement system based on MIR spectroscopy (KNESTEL Technologie und Elektronik GmbH; Hopferbach, Germany). Three different wavelengths in the region of the CO_2_ vibrational mode were used: λ1 = 4.26 μm, λ2 = 4.45 μm, and λ3 = 4.27 μm. The beam of a tunable diode laser was directed at 45° through the corner of the transparent packaging. A two-point calibration with 0% (*v*/*v*) and 40% (*v*/*v*) CO_2_ was carried out on the empty reference trays. All CO_2_ gas concentrations mentioned in the text are to be understood as percent by volume (*v*/*v*).

#### 2.2.3. Destructive Determination of O_2_ and CO_2_ Concentration

To estimate the measuring accuracy of the non-destructive devices, a comparison with a destructive measurement system was made. Therefore, a gas analyzer (MAT1500, A.KRÜSS Optronic GmbH, Hamburg, Germany) with a zirconium dioxide sensor for determining O_2_ and a non-dispersive infrared sensor for determining CO_2_ gas concentration were used. To extract small aliquots (7 mL) of the gas atmosphere, a hollow needle belonging to the measuring device was inserted into the headspace of the packaging via a septum attached to the lid film.

### 2.3. Microbiological Analysis

Samples were prepared for microbiological analysis as already described in Höll et al., 2016 [7], with slight modifications: In a typical analysis, a total of 70 g of chicken strips were weighed in a sample bag (VWR International, Darmstadt, Germany) and homogenized for 120 s with 50 mL Ringer’s solution (Merck KGaA, Darmstadt, Germany) in a stomacher (LabBlender400, Gemini BV, Apeldoorn, Netherlands). A tenfold dilution series of chicken homogenate was prepared with Ringer’s solution. An amount of 100 μL of each dilution was later spread onto the brain heart infusion agar (Carl Roth GmbH & Co. KG, Karlsruhe, Germany) using sterile glass beads. After incubating the plates aerobically at 30 °C for 3 days, the number of colony-forming units on the plates were counted and the units per gram sample (CFU/g) were calculated.

### 2.4. Influence of the Headspace-to Product Ratio

For the selection of an appropriate headspace:meat ratio, a pre-trial was carried out, with two different headspace:meat ratios—6:1 and 3:1. For this purpose, 200 g fresh chicken strips (Donautal Geflügelspezialitäten, Bogen, Germany) for the 6:1 and 400 g fresh chicken strips for the 3:1 ratio were weighed in transparent polypropylene trays (ES-Plastic GmbH, Hutthurm, Germany) and sealed with a semiautomatic tray sealer (T250, MULTIVAC Sepp Haggenmüller SE & Co. KG, Wolfertschwenden, Germany) under a modified gas atmosphere of 70% O_2_/30% CO_2_. Afterwards, the samples were stored at 10 °C for 11 days, as a greater change in the gas atmosphere was expected at this temperature. The O_2_ and CO_2_ concentrations were measured with a destructive measuring device (see Section 2.2.3) during storage.

### 2.5. Storage at Different High-Oxygen Atmospheres

#### 2.5.1. Sample Preparation

For the main trials, a total of 400 g of fresh chicken strips were weighed into transparent polypropylene trays and sealed with the semiautomatic tray sealer at two modified gas atmosphere compositions: 70% O_2_/30% CO_2_ or 80% O_2_/20% CO_2_. For each atmosphere, six packages were prepared using lid films with integrated sensor spots as described above. In addition, 44 samples were prepared for each gas atmosphere without integrated sensor materials for the destructive gas concentration measurement as well as for the sensory and microbiological evaluation. Samples were stored at 4 °C and 10 °C. Furthermore, three empty trays were prepared for each temperature and gas combination with sealed-in sensor spots to monitor the concentrations of O_2_ and CO_2_ without product influence during storage. For destructive monitoring, two empty trays were prepared for each day of measurement and each gas concentration or storage temperature (*n* = 64).

#### 2.5.2. Determination of O_2_ and CO_2_ Gas Concentration

The gas atmosphere of the prepared trays was monitored for 15 days (except for day 3 for the filled trays and days 3, 5, 6, 12, and 13 for the empty trays) via the non-destructive measurement devices for O_2_ and CO_2_ detection. In addition, measurements with the destructive measuring device for O_2_ and CO_2_ detection were carried out on days 0, 1, 4, 6, 8, 11, 13 and 15 to validate the measuring precision of the non-destructive devices for this purpose.

#### 2.5.3. Determination of the Total Viable Count

The total viable count (TVC) was determined for each temperature and gas composition on days 0, 1, 4, 6, 8, 11, 13, and 15 in duplicate.

#### 2.5.4. Sensory Evaluation during Storage

During the storage trial, the samples were investigated by the panelists on days 0, 1, 4, 6, and 8 (4 °C and 10 °C), and on days 11 and 14 for the samples stored at 4 °C. The intensity of the previously specified and trained attributes was evaluated visually and olfactorily on a line scale ranging from 0 to 100 (0 = not perceptible; 100 = strongly perceptible). In addition, it was asked whether the sample was perceived as visually and olfactorily fresh or rotten. This was also classified by a line scale ranging from 0 (fresh) to 100 (rotten). For the evaluation, a sample was defined as no longer acceptable when the average value of the olfactory or visual impression was ≥50.

### 2.6. Statistical Analysis

The drawings and statistical analysis were performed using MS Excel and Origin version 2021 (Origin Lab., Hampton, VA, USA). All data are presented as the arithmetic mean ± standard deviation. For the calculation of the statistical significance, two-sample t-tests (two-tailed) were performed (alpha level 0.05). An analysis of variance was performed to test for equal variances prior to t-test analysis.

## 3. Results and Discussion

### 3.1. Panel Training

The first part of the panel training consisted of a descriptive analysis. The summed attributes of the panel are shown in Table 1 for both the visual and olfactory impressions. For the forced-aged sample, many more attributes were found. This is to be expected, as fresh poultry is rather odorless and characterized by a neutral pink coloration [20]. The named attributes for the forced-aged samples were also mostly described in the literature [20,21,22,23]. The two odors, *buttery* (diacetyl) and *pungent* (acetic acid), which are also linked with the spoilage of poultry [22,23], were not mentioned by the panel. Nevertheless, they were added to the descriptors and were part of the second part of the training. One possible reason for the missing perception of these attributes during the descriptive analysis might be the forcing at room temperature. In this case, the sulfurous impression became very intense, which probably overlapped with the presence of the diacetyl and pungent odor. For the second part of the training, the references showed in Table 2 were chosen for the odor identification test. All panelists were able to correctly match the flavors to their fitting attributes.

### 3.2. Influence of the Headspace to Product Ratio

The measurement of O_2_ and CO_2_ over a storage period of 11 days at 10 °C with two different headspace:meat volume ratios showed very clearly that the headspace volume is decisive for the effect of O_2_ respiration, which was previously described. In Figure 1, it can be seen that the samples with a headspace:meat volume of 3:1 experienced a considerable change in headspace atmosphere, as both O_2_ and CO_2_ reached a concentration between 45 and 50% on day 11. This means that the O_2_ amount decreased and the CO_2_ amount increased, as the starting concentration was 70% O_2_ and 30% CO_2_.

However, the samples with a 6:1 ratio showed hardly any change, with end-concentrations on day 11 of 66.6 ± 0.2% O_2_ and 29.8 ± 0.37% CO_2_. For this reason, the following trials were carried out with a headspace:meat volume ratio of 3:1.

### 3.3. Non-Destructive vs. Destructive Determination of O_2_ and CO_2_ Concentration

To estimate the measuring precision and the suitability of the tested application, a comparison of the non-destructive and destructive measurement systems was carried out and can be seen in Figure 2. Clearly, the measuring points on day 0 and 1 for O_2_ were not in agreement with the destructive measurement. That is due to the fact that the sensor spots were integrated into the packaging lid under atmospheric oxygen partial pressure, and therefore the higher O_2_ concentration of the MAP had to permeate first into the spot area. For the CO_2_ measurement, the values could be recorded directly in real time.

From day 4, the measured values of the non-destructive methods of both O_2_ and CO_2_ were mostly in agreement with the destructive measurements, with individual exceptions, which are explainable by measurement inaccuracies. However, it is also noticeable that a very high standard of deviations were partially present. Presumably, two effects must be distinguished here. If we look at the measuring point on day 11 of graph d, it becomes clear that this must be an effect in which the gas development in different packages was different, since both the O_2_ and the CO_2_ content have a similarly large deviation. Here, the individual development of the gas atmosphere by spoilage—which builds the basis of this research—is again clarified.

### 3.4. Development of the Gas Concentration in Empty and Filled Trays

Since the non-destructive measuring systems had similar accuracies compared to the destructive systems, the following results were obtained using only the non-destructive devices. The results are shown in Figure 3.

For the empty trays, almost no changes in the gas content were detected. The amount of O_2_ and CO_2_ increased and decreased slightly, respectively. In addition, the optical measurement method for O_2_ deviated from the real values at the first two to three measurement points. This was because the sensor spots were sealed into the lid film under atmospheric conditions and the higher O_2_ concentration in the MAP had to permeate into the spot area first, which was previously discussed in Section 3.3.

By comparing the empty and filled trays, the influence of the product was determined. The samples stored at 4 °C and flushed with 70% O_2_ and 30% CO_2_ (Figure 3a) showed the least change, only having a noticeable cross-over on day 13 for O_2_. A “cross-over” was defined as the day when the curve of the respective gas concentration in the filled trays intersects that of the empty trays. This is an indication of a microbiologically induced change in the headspace atmosphere. For CO_2_ content, however, no change in the headspace atmosphere was observed for the filled and empty trays.

The sample stored at 10 °C (Figure 3b) showed significant greater effects. For these samples, a cross-over for O_2_ was detected on day 5, and for CO_2_ was detected one day earlier. For both gases, the first significant deviation between empty and filled trays was present two days after having the cross-over (O_2_: day 7; CO_2_: day 6). Afterwards, a rapid decrease in the O_2_ and a similar increase in the CO_2_ concentration were visible. For both, the O_2_ and the CO_2_ measurements of the filled trays, high standard deviations were visible from day 11 on. As already described in the prior chapter, this was an effect induced by the individual development of the microbiota.

For the 80% O_2_/20% CO_2_ MAP samples stored at 4 °C (Figure 3c), a significant deviation was noted between the filled and empty trays for the CO_2_ measurement from days 12 to 15. However, O_2_ did not show a statistically significant deviation. The cross-over was on day 12 and then the O_2_ content of the filled package decreased steadily until day 15. Samples stored at 10 °C showed the earliest deviation from the empty trays, with a significant change at day 5 for CO_2_ and day 6 for O_2_. The cross-over was on day 4 (CO_2_) and 5 (O_2_), similar to the samples packed at lower oxygen amounts. The O_2_ decrease and CO_2_ increase was even faster, with values almost reaching 100% CO_2_, and totally respired O_2_ on day 15.

The results clearly indicate that a difference in microbiological spoilage was present. However, the fact that the samples stored at 4 °C showed such a low change in gas concentration stands in contrast to the underlying study of Höll et al., 2016 [7]. Herbert et al. showed more similar results in their study, published in 2015. In that study, the O_2_ decrease and CO_2_ increase were comparable for the 4 °C samples, packed under 70% O_2_/30% CO_2_ MAP. The chicken stored at 10 °C had already reached its cross-over at day 2, and the O_2_ decreased to a value of 20%, while the CO_2_ increased to 60% on day 5 [8]. These results raise the question of whether the overall degree of microbiological spoilage has a lesser impact, and the composition of the microbiome may play a much greater role. To further answer this, the next chapter shows the determined TVCs.

### 3.5. Microbiological Growth

All samples had a similar starting value of the total viable count of approximately 10^4^ CFU/g (see Table 3 and Figure 4). The samples that were stored at 10 °C showed a faster and more immediate growth and reached the defined critical value of 10^7^ CFU/g [11] after about 3 (80/20) or 4 days (70/30). The samples that were stored at 4 °C grew significantly slower and reached this value after about 6 (80/20) or 7 (70/30) days. At the end of storage, all the samples were well above the critical limit. The highest value of >10^10^ CFU/g was reached by the sample packed with 80% O_2_ and 20% CO_2_, which was stored at 10 °C. However, the samples stored at 4 °C also reached final values of >10^9^ CFU/g.

Comparing these results with the literature, only partially similar results are evident. Höll et al., for example, observed a different growth of the TVC during their 2016 study, although they chose the same storage conditions (80% O_2_/20% CO_2_ at 4 and 10 °C) and had similar starting values of TVC. For the samples stored at 4 °C, they reached the critical value for the first time at day 10; for the samples stored at 10 °C, it was reached on day 6 [7]. Rossaint et al. also showed a lower increase in their TVC at 4 °C [20]. However, other studies showed results more similar to ours, reaching critical values between day 4 and 6 [8,21,22] for the samples stored at 4 °C, and on day 2 for the samples stored at 10 °C [8].

These results show very clearly that the type of microorganisms is decisive for O_2_ consumption and CO_2_ generation, not necessarily the absolute number of microorganisms. This fact may present difficulties for the prediction of shelf life.

### 3.6. Sensory Evaluation

The sensory panel was able to determine changes for the visual and olfactory attributes that were defined in Table 2. In Figure 5, the sensory evaluation of the samples stored at 4 °C are shown for both gas concentrations. Aside from the visual descriptor, *gloss*, all attributes increased during the storage time. For the visual evaluation, the *overall impression* (fresh/rotten), *slime* (not present/present) and *color* (pink/grey) increased the most, all signs of spoilage. For the orthonasal evaluation, the *overall impression* and the intensity of the *odor* showed the highest change. The descriptor *buttery* also noticeably increased. Except for the descriptor *gloss*, all attributes were still below 50 scores, prior to the end of the shelf life. This becomes even clearer when looking at Figure 5c,d.

For both gas concentrations, the overall olfactory impression reached the limit of 50 scores clearly after the critical value of 10^7^ CFU/g was achieved. More specifically, the sample packed with a MAP of 70% O_2_/30% CO_2_ was classified as rotten only on the last day of evaluation; for the sample packed with 80% O_2_/20% CO_2_, olfactory spoilage was reached on day 11. Poultry stored at 10 °C (see Figure 6) showed a very similar progression to the 4 °C stored sample but at a much faster rate. However, some attributes were even more evident than those in the samples stored at 4 °C. For the visual evaluation, the samples were classified as more *greyish* and *greenish* on day 8. For the olfactory evaluation, the descriptors *buttery* and *cheesy*/*rancid* were slightly higher than the 4 °C samples on day 14. However, again, by reaching the critical microbiological limit value, the impressions were below the 50 scores value and therefore, visual and olfactory were rated as still *fresh*.

Table 4 compares the evaluation of the sensory attributes more precisely. It is also shown which VOC is dominant or typical for the descriptor on high-oxygen-stored poultry and which microorganisms are often responsible for the emitted substances. With respect to this, it should be said that almost all meat spoilage organisms can form the typical spoilage aroma and it is assumed, that the formation of VOCs can be a result of the interactions between different spoilage bacteria, which makes it difficult to assign specific odors to a single species [24]. However, we focused on the involvement of *Brochothrix thermosphacta*, lactic acid bacteria (LAB) (e.g., *Carnobacterium* ssp., *Leuconostoc* ssp., *Lactobacillus* ssp.) and *Pseudomonas* ssp. because, based on previous studies, we assumed that these were the main spoilage bacteria in our chicken samples [7,22,25,26].

All samples showed a noticeable change in meat *color*, changing from *pink* to *grey*. For beef, this effect is related to low oxygen levels, as red oxymyoglobin turns into metmyoglobin, which is known to appear as grey–brown [27]. However, this mechanism cannot be the cause for the color change observed in poultry, as several studies have shown that lower levels of oxygen actually result in a more stable color in fresh poultry [28]. In 2017, Franke et al. also showed that storing poultry in two different oxygen-rich atmospheres resulted in similar greying effects, which is consistent with our experiments [22]. From our point of view, the greying must be an oxidation effect initiated by several microorganisms.

*Gloss* seems not to be an attribute that can be linked with the poultry quality, as no change was observed by the panel.

The *green discoloration* was more present for the samples stored at 10 °C at the end of the storage, compared to the samples stored at 4 °C. However, both samples showed a significant increase during this observation. The greening of food of animal origin is known to be induced by several microorganisms [29,30]; however, several studies were able to connect *Pseudomonas* with this phenomenon as a result of the formation of sulfur compounds [23,31,32,33]. Since the *green discoloration* was clearly increased in the descriptive analysis (see Section 3.1) and a connection with the *rotten egg* attribute was also seen here, this assumption is supported once again.

In general, *slime* formation is not a question of which microorganisms dominate, but how large the microbial count is, as the breakdown of the tissue begins at 10^8^ CFUg^−1^, in other words, after the sample is microbiologically spoiled [34]. This is also consistent with the panel’s perception at the time that the critical limit was reached for most of the samples, as the slime formation clearly already increased in this regard.foods-11-00592-t004_Table 4Table 4Sensory evaluation of the visual and olfactory descriptors under different storage conditions from day 0 to the last day of sensory evaluation (4 °C: Day 14, 10 °C: Day 8). Each + stands for an increase of 10 scores on the line scale, each - stands for a decrease of 10 scores, 0 stands for a change <10 scores. * Responsible VOCs and spoilage bacteria regarding the literature-based assumptions for the tested descriptors, e.g., lactic acid bacteria (LAB).DescriptorDevelopment by Storage atVOCs *Spoilage Bacteria *4 °C10 °C70/3080/2070/3080/20Color+++++++++++++++++Not knownUnclearGloss-00-Not knownNot knownGreen discoloration++++++++++Sulfur compounds [33]*Pseudomonas* ssp., *LAB* [23,30,31,32]Slime++++++++++++++-Diverse [29,34]Buttery+++++++++++++++++2,3-butanedione (diacetyl) [35]*LAB* [25,32] *B. thermosphacta* [29]Fruity+++++Ethyl esters [23,35,36], 1-Hexanol [25]*Pseudomonas* ssp. [36]Rotten egg+++++Sulphur compounds [33]*Pseudomonas* ssp. [23,31]Fishy+++++Trimethylamine [37]*Pseudomonas* ssp. [22]Cheesy/Rancid+++++++++++Isovaleric acid; (2- and 3-methyl) butanoic acid [38,39]*LAB* [25] *B. thermosphacta* [32,39]Pungent++++++Lactic acid [40], acetic acid [41], propanoic acid [25]*LAB* [25] *B. thermosphacta* [32,35,41]


The descriptor with the strongest increase—aside from the *overall visual* and *olfactory impression*—was *buttery*, which is linked with the VOC diacetyl (2,3-butanedione). Diacetyl is synthesized by LABs as *Carnobacterium divergens* or *B. thermosphacta* [25,42], but it can be also present for *Pseudomonas* inoculated samples [35]. With the exception of the 70/30 sample stored at 10 °C, the increase in the attribute *buttery* was rated by the panel as equivalent (++++). Therefore, it is very likely that a mixture of LAB and *B. thermosphacta* was responsible for this VOC formation.

Typical VOCs, which are perceived as *fruity*, are ethyl esters—such as ethyl acetate, ethyl butanoate, ethyl 2-hexenoate, etc.—and are very often produced by *Pseudomonas* [23]. As this aroma was more present for the 10 °C stored samples packed at 80% O_2_/20% CO_2_, we assume that these microorganisms were highly present, especially at the end of storage. This also agrees well with the *greening* previously described. However, the evaluation of the attributes *rotten egg* and *fishy* do not totally fit that presumption, as a highest increase was present here for the 4 °C 80/20 samples.

Volatile fatty acids are responsible for a *cheesy* or *rancid* odor. These are often formed by *B. thermosphacta*, but can also be formed during spoilage where high amounts of LABs are present [25,32,39]. This fits very well with our results, as the samples stored at 10 °C, and the 80/20 sample stored at 4 °C showed an increase in these attributes. These samples also showed a decrease in the concentration of O_2_ in the headspace. This is in line with the respiration abilities of these microorganisms, especially with those of *B. thermosphacta* and *Leuconostoc gelidum* [10].

The oxygen consumption by *B. thermosphacta* and LABs for the samples stored at 10 °C can be further underlined by the higher increase in the attribute *pungent* for these samples, as the formation of acetic and lactic acid is linked with these meat spoilers [32,35,40,41].

### 3.7. Correlation of the Results

Table 5 gives an overview of the tested parameters and indicates some observed correlations. “Correlations” are understood as mutual, temporal relationships. The correlations refer to effects (e.g., the change in gas content and microbiological spoilage) that occurred at the same or a similar point in time. Yellow describes a possible association between the cross-over and microbiological spoilage. Orange indicates a correlation between the cross-over and sensory changes. Green indicates a correlation between the change in gas concentrations in the headspace and sensory changes. Grey stands for a correlation of microbiological spoilage and sensory changes.

#### 3.7.1. Correlation of Change in Gas Concentrations with Microbiological Spoilage

As described previously, some studies were carried out that also indicate a correlation between O_2_ respiration and microbiological spoilage for chicken meat [7,9]. However, this study could not completely confirm these findings. For all samples, no significant change in the gas concentration of the headspace was noted when the limit of 10^7^ CFU/g was reached. Microbial spoilage might be visible with the help of the previously described cross-over point for CO_2_ detection; however, this is only valid for the samples that were stored at 10 °C. This point was reached either on the day (70/30) or 1 day after (80/20) the critical value of 10^7^ CFU/g was reached. The suggestion that the cross-over could be an indication of premature spoilage, e.g., due to improper storage, is further underlined in the literature, where chicken samples were also stored at 70% O_2_ and 30% CO_2_ at 10 and 15 °C, and the cross-over of the reference and the packed sample for O_2_ and CO_2_ measured with a destructive device (=empty tray) were consistent in reaching their critical value [8].

A significant change in the headspace gas atmosphere occurred in the samples stored at 10 °C, and also for CO_2_ detection in the 80/20 4 °C samples; however, the TVC was already ≥10^9^ CFU/g in these cases. For the samples stored at 10 °C, a very strong change occurred for both gases in the gas atmosphere after 5 or 6 days. At the end of the storage period, the gas atmosphere completely changed in the headspace. Nevertheless, the total microbiological growth was not very different compared with the 4 °C samples after 15 days. This is a strong indication that the type of microorganisms, and not necessarily the quantity, was crucial for O_2_ consumption. This was confirmed in a study wherein beef was inoculated with different meat-spoiling bacteria. Samples contaminated with *Brochothrix thermospacta* showed significant O_2_ consumption, while samples contaminated with *Carnobacterium divergens* and *Carnobacterium maltaromaticum* showed no consumption at all, even at microbial populations of ≥10^8^/cm^2^ [10]. This shows that it is mainly the individual microbiota that determines O_2_ consumption and CO_2_ production, which makes a shelf-life prediction on this basis difficult. However, the cross-over seems to be a useful tool for identifying inappropriately stored samples, e.g., in case of cold-chain interruption.

#### 3.7.2. Correlation of Change in Gas Concentrations with Sensory Evaluation

There appeared to be a correlation between the gas development and sensory acceptance or descriptors in some cases. For the poultry that was stored at 4 °C at initial gas concentrations of 80% O_2_ and 20% CO_2_, a significant change in CO_2_ was observed on day 12, and the cross-over happened on day 10. This agreed with the *overall olfactory impression* on day 11 when the panel classified the sample as *not acceptable* for the first time. The 4 °C 70/30 sample had its cross-over with O_2_ on day 13 and the first classification as *olfactory spoiled* on day 14. However, the classification on day 11 was slightly below the 50 scores limit, which is why a prediction of shelf life via a determination of the composition of the headspace gases is rather unlikely. The *overall olfactory impression* was probably mostly induced by the descriptor *cheesy*/*rancid* in this case, as both attributes were rated by the panel with an increase of >20 scores compared to day 0 on the same days for each gas composition (80/20: day 11; 70/30: day 14). For the samples stored at 10 °C, however, a very good fit between the gas concentration and the sensory changes was observed. The 80/20 samples experienced their significant change in CO_2_ on day 5 and for O_2_ on day 6. At this time, the *overall olfactory impression* was scored as spoiled by the panel and the *greyish color*, and the descriptors *buttery* and *cheesy*/*rancid* increased by >20 scores. For the 70/30 sample, the gas change occurred one day later; for both gases and the sensory changes, the defined levels were reached on day 8, except for *slime*.

#### 3.7.3. Correlation of Sensory Evaluation with Microbiological Spoilage

Several sensory attributes show a correlation with microbiological spoilage. For the samples stored at 4 °C, the descriptors *slime* and *buttery* increased by >20 scores on day 6 (80/20) or day 7 (70/30), while the critical value of 10^7^ CFUg^−1^ was reached on day 6 for both initial gas compositions. Inappropriately stored samples at 10 °C also showed the *slime* formation when the critical TVC was reached. This fact demonstrates the need for other detection methods, especially for the samples stored at 4 °C, as the non-destructive gas evaluation seems not to be suitable here. The identification of *slime* could be possible using other spectroscopic analysis methods such as absorption spectroscopy in the infrared or Raman spectroscopy on the surface of the sample. Another possibility would be the use of hyperspectral imaging. All of these are useable as non-destructive methods for already packed products as well [12].

### 3.8. Influence of the Microbiome

These results allow some conclusions to be drawn about the type of spoilage microorganisms, with some limitations. Franke et al. (2017) showed that chicken breasts packed at high O_2_ and stored at 4 °C were mainly populated with *B. thermospacta,* and developed *Carnobacteria* sp. and *Pseudomonas* sp., mainly under MAP with lower CO_2_ concentrations (≤15%) [22]. Because *Pseudomonas* sp. are responsible for the formation of VOCs [20], this agreed well with the earlier sensory spoilage of the 80/20 sample, compared with that of the 70/30 sample. However, O_2_ was hardly respired at this point, which could be because of the small population of *B. thermospacta;* the lack of heme, compared to beef; or because of the low temperature [9,10]. The influence of storage temperature was visible, especially with respect to sensory evaluation and gas development. In addition, this was also observed by Höll et al. (2016). They had a mixed microbiota at the beginning of storage. Later, at 4 °C, a mixture of *B. thermospacta*, *Pseudomonas* sp., and *Carnobacteria* sp. grew, and at 10 °C, the microbiota mainly consisted of *Pseudomonas* sp. and *Serratia* sp. at the end of storage period. After 4–8 days, *B. thermospacta* was present, which likely favored O_2_ consumption. Then, VOC-forming *Pseudomonas* sp. could grow [7]. This effect was likely the same as that observed in this study.

In conclusion, the composition of the microbiome in this work can only be estimated based on the gas change in the headspace and the VOCs produced. So, a reliable statement can be made and the clarification of the species would have to be carried out in the future, e.g., using MALDI-TOF MS.

## 4. Conclusions

This study showed that the presented non-destructive measurement systems can monitor the O_2_ and CO_2_ concentrations in the headspace of a package in real time and showed little to no deviations to the destructive methods in which gas samples were extracted from the headspace. The measurement of the CO_2_ with the infrared-based systems showed no problems. The observed variations in headspace gas composition could be attributed to individual compositions of the microbiota on the samples. The systems, however, cannot be used to predict the shelf life of high-O_2_-packed poultry stored under regular conditions and the resulting regular decay mechanisms. For the detection of premature microbial spoilage, however, e.g., due to contamination or an interruption in the cold chain, CO_2_ detection might be especially useful because a significant change in the CO_2_ fraction of the headspace could be observed before sensory spoilage, and the defined cross-over was correlated with reaching the critical value of 10^7^ CFU/g. The predominant sensory changes identified by the successfully trained panel were *overall olfactory* and *visual impression*, the formation of *slime*, *color* change from *pink* to *grey*, and the formation of the odors *buttery* and *cheesy*/*rancid*. However, a correlation with the microbiological shelf life (10^7^ CFU/g) was only possible with the formation of *slime* (4 °C and 10 °C) and *buttery* (4 °C). As the formation of VOCs also strongly depends on the microbiome, the detection of diacetyl (=*buttery*) is not a recommendable approach in our view. This is particularly true because the detection of individual substances using simple indicators can be disturbed by the presence of other substances. The optical detection of *slime* could instead be more promising. In addition, the influence of the heme concentration should be clarified by experiments with beef in the future, as this seems to have a main role in O_2_ consumption. Another possible application for the technologies might be the detection of leakages in packages or process control for MAP production lines. For this purpose, further investigations should be carried out on the measurement accuracy of the two non-destructive methods.

## Figures and Tables

**Figure 1 foods-11-00592-f001:**
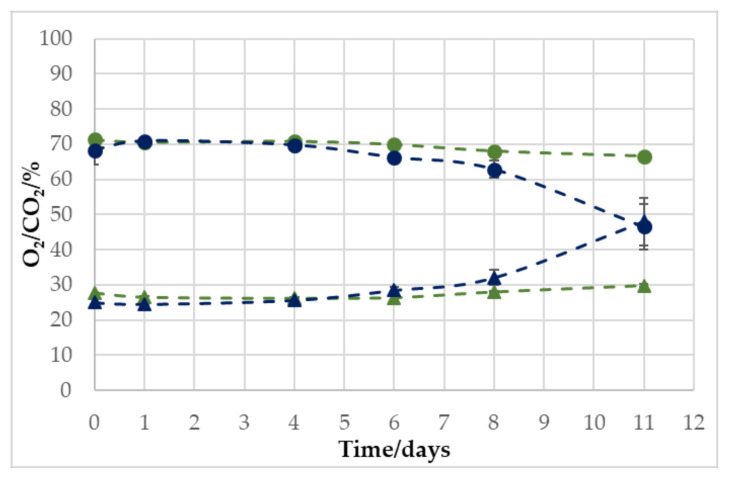
Development of the relative volume concentration of O_2_ (●/●) and CO_2_ (▲/▲) for a headspace:meat ratio of 6:1 (●/▲) and 3:1 (●/▲) at a storage temperature of 10 °C for 11 days (*n* = 2 for each measuring point).

**Figure 2 foods-11-00592-f002:**
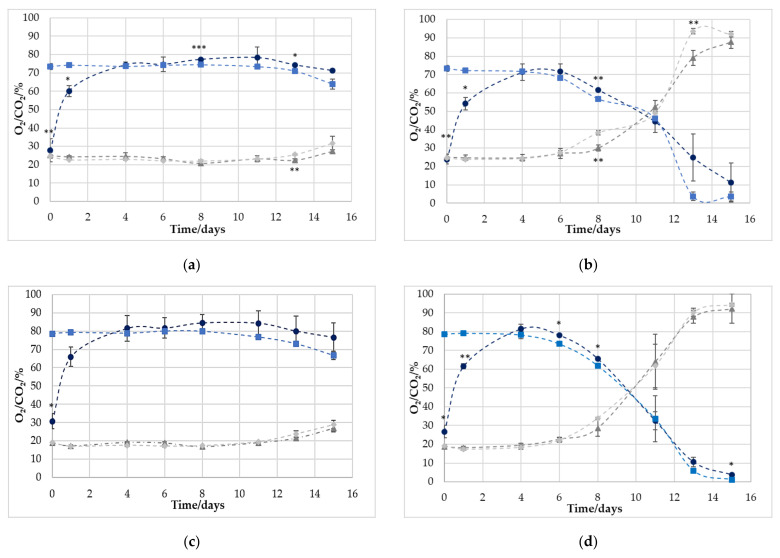
Comparison between the destructive (■/◆) and non-destructive (●/▲) measurement devices for the investigation of O_2_ (■/●) and CO_2_ (◆/▲) during the storage of poultry over 15 days under different storage conditions: (**a**) 70% O_2_/30% CO_2_ 4 °C (**b**) 70% O_2_/30% CO_2_ 10 °C (**c**) 80% O_2_/20% CO_2_ 4 °C (**d**) 80% O_2_/20% CO_2_ 10 °C. Indices indicate a significant difference between the curves measured with the destructive and non-destructive measurement devices: * *p* < 0.05, ** *p* < 0.01, and *** *p* < 0.001.

**Figure 3 foods-11-00592-f003:**
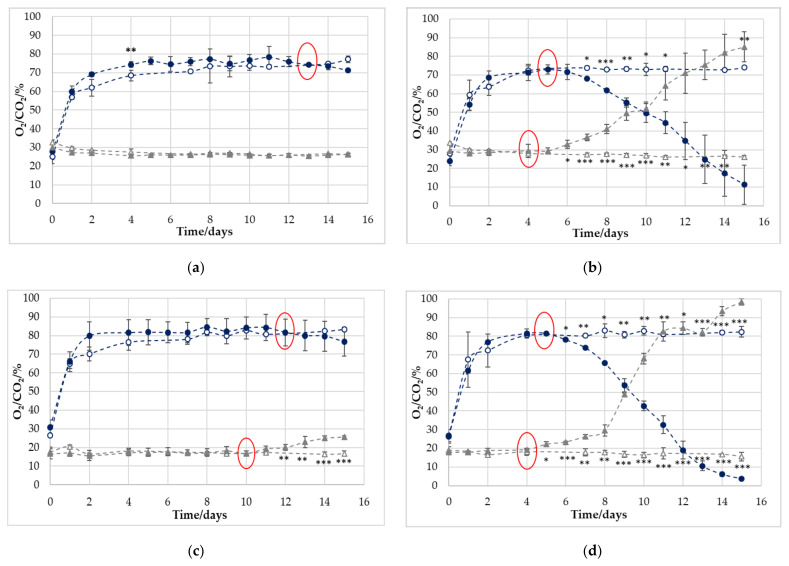
Development of O_2_ (○/●) and CO_2_ (△/▲) under different storage conditions over 15 days in trays with (●/▲) and without poultry (○/△): (**a**) 70% O_2_/30% CO_2_ 4 °C (**b**) 70% O_2_/30% CO_2_ 10 °C (**c**) 80% O_2_/20% CO_2_ 4 °C (**d**) 80% O_2_/20% CO_2_ 10 °C. Indices indicate a significant difference between the curves with and without poultry: * *p* < 0.05, ** *p* < 0.01, and *** *p* < 0.001. The red circles mark the point when the curve of the respective gas concentration in the filled trays intersects that for the empty trays (cross-over), which indicates a microbiologically induced change in the headspace atmosphere.

**Figure 4 foods-11-00592-f004:**
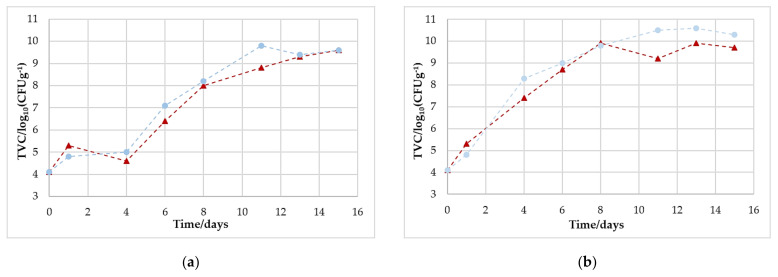
Development of the total viable count (TVC) for poultry stored under different storage conditions over 15 days in trays with initial gas concentrations of 70% O_2_/30% CO_2_ (▲) and 80% O_2_/20% CO_2_ (●) at (**a**) 4 °C and (**b**) 10 °C.

**Figure 5 foods-11-00592-f005:**
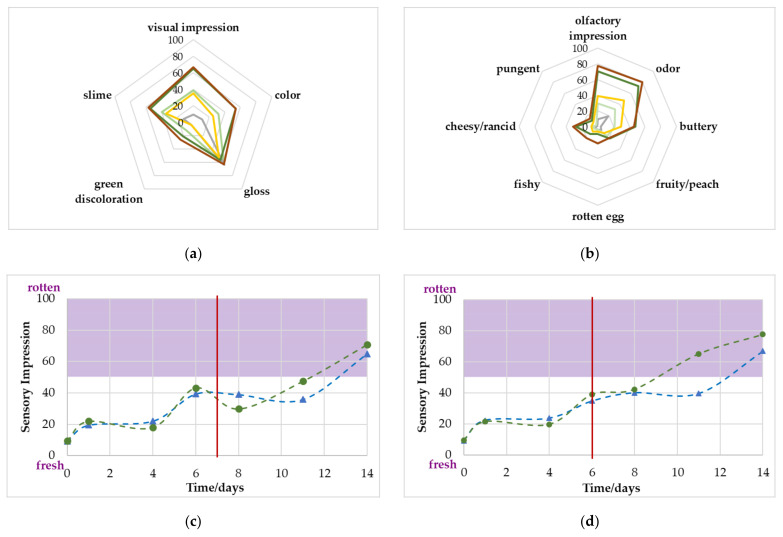
(**a**,**b**): Sensory evaluation of the poultry stored at 4 °C by the panel with the chosen descriptors for (**a**) visual and (**b**) olfactory attributes on day 0 (grey line), after reaching the critical value of 10^7^ CFU/g (70% O_2_/30% CO_2_: light green; 80% O_2_/20% CO_2_: yellow) and on day 14 (70% O_2_/30% CO_2_: dark green; 80% O_2_/20% CO_2_: brown). (**c**,**d**): Change in the overall visual (▲) and olfactory (●) impression during the storage time: (**c**) 70% O_2_/30% CO_2_ 4 °C and (**d**) 80% O_2_/20% CO_2_ 4 °C. The red line marks the point when the microbiological limit value was achieved. The purple area indicates the previously defined sensory limit of 50 scores.

**Figure 6 foods-11-00592-f006:**
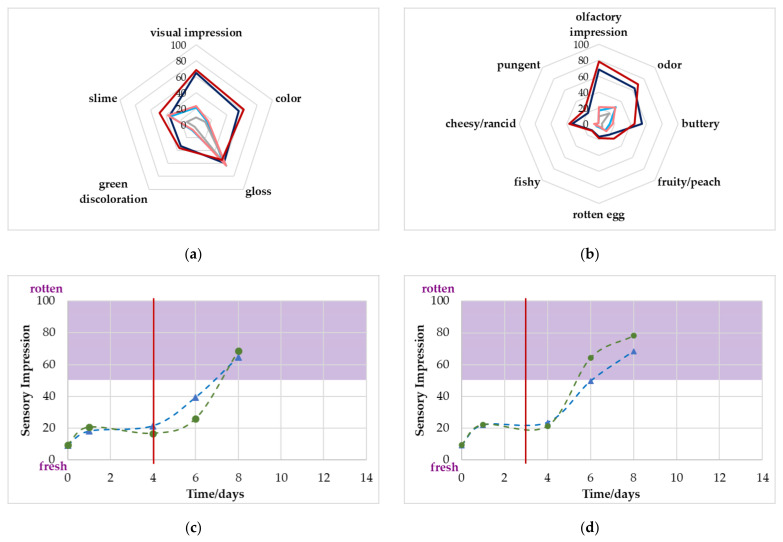
(**a**,**b**): Sensory evaluation of poultry stored at 10 °C by the panel with the chosen descriptors for (**a**) visual and (**b**) olfactory attributes on day 0 (grey line), after reaching the critical value of 10^7^ CFU/g (70% O_2_/30% CO_2_: light blue; 80% O_2_/20% CO_2_: pink) and on day 8 (70% O_2_/30% CO_2_: dark blue; 80% O_2_/20% CO_2_: red). (**c**,**d**): Change in the overall visual (▲) and olfactory (●) impression during the storage time: (**c**) 70% O_2_/30% CO_2_ 10 °C and (**d**) 80% O_2_/20% CO_2_ 10 °C. The red line marks the point when the microbiological limit value was achieved. The purple area indicates the previously defined sensory limit of 50 scores.

**Table 1 foods-11-00592-t001:** Results of the descriptive analysis for the visual and olfactory attributes named by the panel for a fresh and forced-aged (4 days at 23 °C) poultry sample.

Visual Impression				Olfactory Impression			
Fresh Poultry		Forced-Aged Poultry		Fresh Poultry		Forced-Aged Poultry	
Fresh	11	Grey	11	Fresh	10	Rotten egg/sulfurous	15
Pink	10	Green	8	Neutral	8	Rotten	9
Salmon-colored	3	Slime	6	Poultry typical	2	Rancid	7
Solid shape	2	Glossy	6	Weak	1	Fishy	6
Skin-colored	1	Weeping/wet	5			Cheesy	5
Appealing	1	Rotten	3			Sweet	3
		Brown	2			Fruity	2
		Yellowish	1			Acidic	2
		Mushy	1			Ammonia	1
						Flowery	1
						Lemony	1
						Drain-like	1

**Table 2 foods-11-00592-t002:** Chosen descriptors for the sensory evaluation, based on the descriptive analysis of the panel training. The references and scale for the olfactory impression were also used for the second part of the training—the odor identification test.

Descriptor	Scale	Reference
**Visual Impression**		
Overall impression	0 (fresh)–100 (rotten)	Images fresh and forced-aged sample
Color	0 (pink)–100 (grey)	Color chart
Green discoloration	0 (none)–100 (clear)	Color chart
Slime	0 (not present)–100 (present)	-
Gloss	0 (weak)–100 (strong)	Images fresh and forced-aged sample
**Olfactory Impression**		
Overall impression	0 (fresh)–100 (rotten)	-
Odor	0 (neutral)–100 (intense)	Water
Buttery	0 (not perceptible)–100 (clearly perceptible)	Diacetyl
Fruity/peach	0 (not perceptible)–100 (clearly perceptible)	Gamma-decalactone
Rotten egg	0 (not perceptible)–100 (clearly perceptible)	Sodium sulfide
Fishy	0 (not perceptible)–100 (clearly perceptible)	Cis-4-heptenal
Cheesy/rancid	0 (not perceptible)–100 (clearly perceptible)	Butanoic acid
Pungent	0 (not perceptible)–100 (clearly perceptible)	Acetic acid

**Table 3 foods-11-00592-t003:** Total viable count (TVC) for the poultry on days 0 and 15 for each storage condition (*n* = 4) and the day the critical limit of 10^7^ CFUg^−1^ was reached (end of shelf life).

	Day 0	Day 15	Shelf Life Expired
**80/20 4 °C**	1.36 × 10^4^ CFUg^−1^	4.00 × 10^9^ CFUg^−1^	Day 6
**70/30 4 °C**	1.27 × 10^4^ CFUg^−1^	4.29 × 10^9^ CFUg^−1^	Day 7
**80/20 10 °C**	1.36 × 10^4^ CFUg^−1^	2.19 × 10^10^ CFUg^−1^	Day 3
**70/30 10 °C**	1.27 × 10^4^ CFUg^−1^	4.47 × 10^9^ CFUg^−1^	Day 4

**Table 5 foods-11-00592-t005:** Observed correlations between the tested parameters. *p* ≤ 0.05 represents the first day where the difference between empty and filled trays was significant with *p* ≤ 0.05, after the cross-over was reached. “Microbiologically spoiled” indicates a TVC of 10^7^ CFUg^−1^ and “sensory changes” shows the panel classification of the sensory panel with: >50 scores for the overall olfactory impression (“olfactory spoiled”) and an increase of >20 scores in the descriptors *slime*, *color*, *buttery* and *cheesy*. A description of the color marking of the correlations can be found in the previous text.

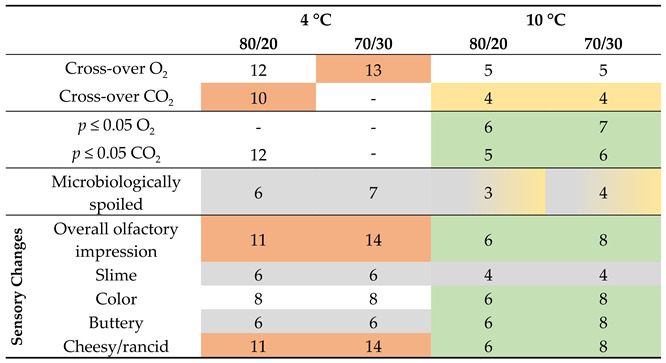

## Data Availability

Not applicable.

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
