# Peer review of "Non-Destructive Measuring Systems for the Evaluation of High Oxygen Stored Poultry: Development of Headspace Gas Composition, Sensory and Microbiological Spoilage"

_foods, 2022, doi:10.3390/foods11040592_

Round 1
Reviewer 1 Report
The paper is focused on the evaluation of the shelf life of high oxygen packaged chicken by no invasive methods (quenching fluorescence and MIR). I think the paper is correctly designed, and the results and discussion section very interesting. I miss a higher work in the determination of microbial groups present, not only total viable count, but also LAB, pseudomonads or even yeast during the storage, as well as the correlation with O2/CO2 changes. As the authors mention not only the absolute number is important but also the type or genera involved. I have some minor comments:
Line 170: Which was the appropriate headspace/meat ratio? I red later that authors tested two ratios. It would be helpful to include this information in material and methods section.
Line 328: in my opinion the microbial behavior could be presented in 2 graphics, one for each storage temperature, to have a better understanding of the shelf life evolution.
Line 384: In my opinion other spoilage lactic acid apart from Carnobacterium should be considered as Leuconostoc or Weissella viridescens in table 4. It is difficult when there is a mixed spoilage microflora to attribute specific odors to a single microbial species. I would recommend considering the spoilage bacteria in a general way.
Author Response
Dear Reviewer,
Please see the attachment.
Kind regards,
Jasmin Dold

Reviewer 2 Report
I read this paper with great interest. It is a nice piece of work.
The only great issue is the lack of good statistical analyses. There is no clear data analyses in this paper and this represents a big drawback that should be clearly improved by the authors. For example, they didn't use any robust tool to handle their data and only mentioned Excel.
The authors mentioned correlations but not able to see them.
I would suggest to the authors to apply clear analysis and define their model. Also, a multivariate analysis such as Principal Component Analysis would be useful in this study.
Indeed, the results and discussions sections should be corrected accordingly and improved.
Author Response
Dear Reviewer,
Please see the attachment.
Kind regards
Jasmin Dold
